

# Symmetries, safety, and self-supervision

**Barry M. Dillon[1], Gregor Kasieczka[2], Hans Olischläger[1],**
**Tilman Plehn[1], Peter Sorrenson[1,3] and Lorenz Vogel[1]**

**1** Institut für Theoretische Physik, Universität Heidelberg, Germany
**2** Institut für Experimentalphysik, Universität Hamburg, Germany
**3** Heidelberg Collaboratory for Image Processing, Universität Heidelberg, Germany

## Abstract

Collider searches face the challenge of defining a representation of high-dimensional data such that physical symmetries are manifest, the discriminating features are retained, and the choice of representation is new-physics agnostic. We introduce JetCLR to solve the mapping from low-level data to optimized observables through self-supervised contrastive learning. As an example, we construct a data representation for top and QCD jets using a permutation-invariant transformer-encoder network and visualize its symmetry properties. We compare the JetCLR representation with alternative representations using linear classifier tests and find it to work quite well.



## 1 Introduction

Symmetries [1] form the core of the fundamental description, phenomenological techniques, and experimental analyses in particle physics. LHC physics is defined by the symmetry structure of LHC data, from the detector geometry to the relativistic space-time symmetries and

local gauge symmetries defining the underlying theory, and to new physics motivations like supersymmetry. Any new approach to LHC physics, including applications of machine learning, has to be seen in the context of symmetries eventually [2–5].

Modern machine learning (ML) has spurred the development of techniques which can, among other benefits, boost the development of high-level observables. We typically train a neural network to distinguish between different physical processes either based on high-dimensional, low-level data or on the corresponding Monte Carlo simulations. The resulting classifier can be viewed as a high-level observable for a given analysis. If the dataset can be understood through first-principle simulations and is large enough to train the networks, this observable will be optimized for the respective task, but lack theoretical calculability. As long as the observable is calibrated and the systematic errors are understood, this lack of calculability is not a barrier for supervised classification.

This agnostic approach works well for supervised analyses, but it is not clear how it can be expanded to unsupervised analyses, like an anomaly search [6–10], a generalized side band analysis [11, 12], or a generalized model hypothesis [13]. For this purpose, we propose to replace a limited number of high-level observables by a high-dimensional representation, and replace full control over all possible physics processes with a structure driven by symmetries and fundamental theory.

The standard application driving ML methods in LHC physics is jets, a fertile ground for supervised and unsupervised techniques. The most common jet representation is jet images [14–19], a high-dimensional representation defined in rapidity vs. azimuthal angle, observables inspired by Lorentz transformations. Jet images typically include a preprocessing step exploiting their rotation symmetry. Alternative symmetry-inspired jet representations include permutation-invariant graphs [20–24], trees [25–28], the Lund plane [29,30], Lorentz-inspired networks [31–35], Deep-Sets networks [36], or energy flow polynomials (EFPs) [37], a calculable basis with a notion of infrared and collinear safety.

Combining unsupervised learning and symmetries we define jet observables using contrastive learning of representations (CLR) [38]. Our key idea is to frame the mapping between the jet constituents' phase space and a representation space as an optimization task with a contrastive loss function, *designed such that the representation space will be invariant to pre-defined symmetries and retains discriminative power*. The training is self-supervised in view of the network's discriminative power, because the optimization never uses truth labels for the jets. For the mapping of physics and representation spaces we employ a transformer-encoder network [39–41]. In addition to its built-in permutation symmetry we implement rotation and translation symmetries, as well as soft and collinear safety augmentations. To benchmark JetCLR we use a standard test in the ML community, the so-called linear classifier test (LCT). For this test a linear network is trained to classify between different processes, quantifying how well classes can be separated by a linear cut in representation space.

We start by introducing contrastive learning in Section 2. We then construct our JetCLR tool using a set of symmetries and augmentations in Section 3. In Section 4 we visualize the invariances of the JetCLR representation and study its performance using a linear classifier. Different such classifiers are discussed in the Appendix.

## 2 Contrastive learning

The goal of our network is to define a mapping between the jet constituents and a representation space,

$$f : \mathcal{J} \to \mathcal{R}, \tag{1}$$

which is both,

1. invariant to symmetries and theory-driven augmentations, and

2. discriminative within the dataset it is optimized on.

We do this using contrastive learning. Positive and negative pairs of jets are generated using the training data and theory-driven augmentations of the training data. A neural network is then used to optimize the mapping $f$ with respect to a loss function taking these positive and negative pairs as inputs. A detailed description of the procedure is given below.

We work with the top-tagging dataset [6, 31, 42, 43], where the jets are simulated with PYTHIA 8.2 [44] (default tune) using a center-of-mass energy of 14 TeV and ignoring pile-up and multi-parton interactions (MPI). In a practical experimental setting some procedure to remove these effects would be required, such as jet grooming [45–47], however to separate the tasks of jet tagging and pile-up/MPI removal these effects are ignored here. The simulation models the production of the partons and subsequent showering and hadronisation. DELPHES [48] provides a fast detector simulation with the default ATLAS detector card, such that the constituents measured in the jet correspond to energy deposits in the calorimeters. The jets are defined through anti-$k_\mathrm{T}$ algorithm [49, 50] in FASTJET [51] with a radius of $R = 0.8$. For each event we keep the leading jet, provided

$$p_\mathrm{T} = 550 \ldots 650 \,\mathrm{GeV} \qquad \text{and} \qquad |\eta| < 2 \,. \tag{2}$$

This narrow $p_\mathrm{T}$-range induces the most distinctive feature in the jets, a finite geometric distance between the top decay products in the $\eta$–$\phi$ plane, whereas for QCD jets the average activity continuously drops away from the hardest constituent. The top jets are required to be matched to a parton-level top and all parton-level decay products to lay within the jet radius. The jet constituents are defined using the DELPHES energy-flow algorithm, with the leading 200 constituents from each jet kept for the analysis. Particle-ID and tracking information are not included.

If we assume all jet constituents to be massless, each jet $x_i$ is defined by its constituent coordinates,

$$x_i = \{(p_\mathrm{T}, \eta, \phi)_k\} \qquad \text{with} \qquad k = 1, \ldots, n_C \,, \tag{3}$$

so the jet phase space $\mathcal{J}$ has dimension $3n_C$. For the training, we first sample a batch of jets $\{x_i\}$ from the dataset and apply a set of symmetry-inspired augmentations to each jet to generate an augmented batch $\{x_i'\}$. Pairs of original and augmented jets are defined as

$$\text{positive pairs:} \qquad \{(x_i, x_i')\} \,, \tag{4a}$$

$$\text{negative pairs:} \qquad \{(x_i, x_j)\} \cup \{(x_i, x_j')\} \quad \text{for} \quad i \neq j \,. \tag{4b}$$

The goal of the network training is to map positive pairs close together in the representation space $\mathcal{R}$ and negative pairs far apart. This way, positive pairs are used to impose invariances under symmetry transformations or theory augmentations of the jets in $\mathcal{R}$, while the negative pairs are used to ensure that the representation retains discriminative power within the dataset. Truth labels indicating if the jets are QCD or top are never used in the optimization.

**Loss function**

The mapping of Eq. (1) defines the network outputs $z_i$ and $z_i'$, each of them vectors describing jets in $\mathbb{R}^{\dim(z)}$. The actual representation, however, is given by $f(x_i) = z_i/|z_i|$ and $f(x_i') = z_i'/|z_i'|$, which means it is defined on a unit hypersphere

$$\mathcal{R} = S^{\dim(z)-1} \,. \tag{5}$$

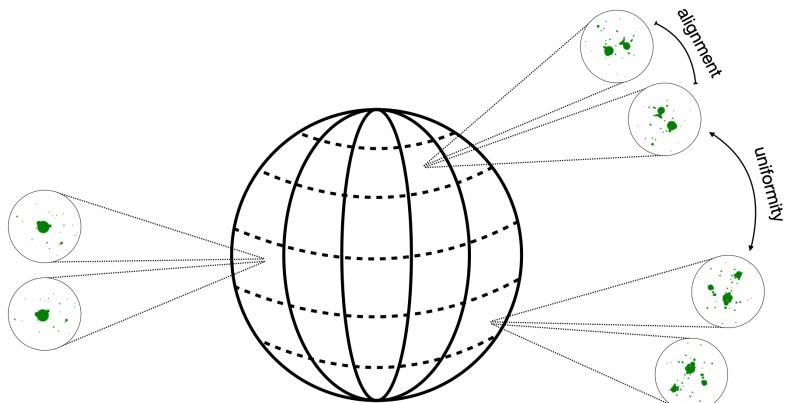

Figure 1: Illustration of the uniformity and alignment concepts behind the contrastive learning framework.

On this sphere we define the similarity between two jets as [38]

$$s(z_i, z_j) = \frac{z_i \cdot z_j}{|z_i||z_j|} = \cos \theta_{ij}, \tag{6}$$

with $\theta_{ij}$ being the angle between the jets in $\mathcal{R}$. The contrastive loss for a positive pair of jets is defined in terms of this distance as

$$\mathcal{L}_i = -\log \frac{e^{s(z_i, z_i')/\tau}}{\sum_{j \neq i \in \text{batch}} \left[ e^{s(z_i, z_j)/\tau} + e^{s(z_i, z_j')/\tau} \right]}, \tag{7}$$

and the total loss is computed as the mean over all positive pairs in the batch. Because the positive pairs appear in the numerator, while the negative pairs contribute to the denominator, the loss decreases when the distance between positive pairs becomes smaller and when the distance between negative pairs becomes larger. The hyper-parameter $\tau$ is referred to as the temperature and controls the relative influence of positive pairs and negative pairs. The cosine similarity in Eq. (6) is not a proper distance metric, but we can define an angular distance as $d(z_i, z_j) = \theta_{ij}/\pi = 0 \dots 1$, such that it satisfies the triangle inequality.

**Uniformity vs. alignment**

The contrastive loss can be understood in terms of uniformity versus alignment on the unit hypersphere defining $\mathcal{R}$, illustrated in Fig. 1. The numerator of Eq. (7), describing the positive pairs, is minimal when all jets and their augmented counterparts are mapped to the same point, $s(z_i, z_i') = 1$. On a hypersphere, the negative pairs cannot be pushed infinitely far apart, as would be possible in $\mathbb{R}^{\dim(z)}$, so the corresponding loss is minimal when the jets are uniformly distributed on the hypersphere. We can measure uniformity and alignment through [52]

$$\mathcal{L}_{\text{align}} = \frac{1}{N_{\text{batch}}} \sum_{i \in \text{batch}} s(z_i, z_i'), \tag{8a}$$

$$\mathcal{L}_{\text{uniform}} = \frac{1}{N_{\text{batch}}} \sum_{i \in \text{batch}} \log \sum_{j \neq i} \left[ e^{-s(z_i, z_j)} + e^{-s(z_i, z_j')} \right]. \tag{8b}$$

While the alignment function has the trivial solution where all jets and all augmented jets are mapped to the same point, the uniformity function does not have such a solution. To map the

jets to a uniform distribution in a high-dimensional space, the mapping must learn features of the jets to discriminate between them and map them to different points. In this respect the choice of distance in Eq. (6) is crucial to the construction of this method. Alternative distance measures could in principle be used, however they must lead to the same uniformity behaviour during optimization. Uniformity alone is a sufficient optimization task to obtain a representation with discriminative power. The additional alignment condition develops a mapping to $\mathcal{R}$, which focuses on the invariance with respect to augmentations and symmetries. The combined contrastive learning will not find representations which are perfectly aligned or perfectly uniform.

**Symmetries and augmentations**

The mapping to the representation space $\mathcal{R}$ is optimized to be approximately invariant to pre-defined symmetry transformations and data augmentations. Before applying symmetry transformations and augmentations in the contrastive learning method we center the jets such that the $p_{\mathrm{T}}$-weighted centroid is at the origin in the $\eta$–$\phi$ plane.

Rotations around the jet axes turn out to be a very efficient symmetry we can impose on our representations. In the jet-image representation this is included through preprocessing, where each jet is centered and then rotated such that its principal axis points at 12 o'clock. Energy flow polynomials are rotationally invariant by construction, since they are built from angular distances between the jet constituents. We apply rotations to a batch of jets by rotating each jet through angles sampled from $0 \dots 2\pi$. Such rotations in the $\eta$–$\phi$ plane are not Lorentz transformations and do not preserve the jet mass, but for narrow jets with $R \lesssim 1$ the corrections to the jet mass can be neglected.

As a second symmetry we implement translations in the $\eta$–$\phi$ plane. To do so, all constituents in a jet are shifted by the same random distance, where shifts in each direction are limited to between $-1 \dots 1$ and the distance is different for each jet. This performs better than restricting to smaller shifts. Although we do an initial centring of the jets, we would like the representations to be invariant to shifts, and so only depend on the distances between constituents in the jet.

In addition to (approximate) symmetries, we also employ theory-inspired augmentations. The distinction between the two is much clearer in our physics application than it is in traditional machine learning. Quantum field theory tells us that soft gluon radiation is universal and factorizes from the hard physics in the jet splittings [53,54]. To encode this invariance in $\mathcal{R}$ we augment our jets by smearing the positions of the soft constituents, i.e. by re-sampling the $\eta$ and $\phi$ coordinates of each constituent from a Gaussian distribution centred on the original coordinates,

$$\eta' \sim \mathcal{N}\left(\eta, \frac{\Lambda_{\mathrm{soft}}}{p_{\mathrm{T}}}\right) \qquad \text{and} \qquad \phi' \sim \mathcal{N}\left(\phi, \frac{\Lambda_{\mathrm{soft}}}{p_{\mathrm{T}}}\right), \tag{9}$$

with a $p_{\mathrm{T}}$-suppression in the variance relative to $\Lambda_{\mathrm{soft}} = 100\,\mathrm{MeV}$.

Similar to soft splittings, also collinear splitting lead to divergences in perturbative quantum field theory. In practice, they are removed through the finite angular resolution of a detector, which will not be able to distinguish two constituents with $p_{\mathrm{T},a}$ and $p_{\mathrm{T},b}$ at vanishing separation $\Delta R_{ab} \ll 1$. We introduce collinear augmentations to encode this feature by selecting constituents and splitting them such that the total $p_{\mathrm{T}}$ in an infinitesimal region of the detector is unchanged,

$$p_{\mathrm{T},a} + p_{\mathrm{T},b} = p_{\mathrm{T}}, \qquad \begin{aligned} \eta_a = \eta_b &= \eta, \\ \phi_a = \phi_b &= \phi. \end{aligned} \tag{10}$$

Our soft and collinear augmentations will enforce an approximate IRC-safety in the jet representation. Unlike for instance EFPs, we do not explicitly enforce it through a fixed set of

angular correlations or $p_T$-scalings, but let the contrastive optimization determine the mapping to the representation space.

# 3 JetCLR

The symmetries discussed in the last section leave out one of the key symmetries in jet representations, namely permutation symmetry. This simply means that the definition of a jet observable should not rely on any specific ordering of the jet constituents. While it seems obvious, many machine-learning tools in the literature input the jet constituent data to neural networks with a fixed ordering, for example ordered by $p_T$. Several other approaches developed in machine-learning applications to high-energy physics have also addressed the permutation symmetry of particles [21, 36, 37, 40, 55]. We will include it through the transformer architecture, which shares similarities with the Deep-Sets architecture used in Ref. [36], mapping jet phase space to representation space. The combination of contrastive loss and a permutation-invariant network architecture defines our JetCLR concept.

**Attention mechanism**

The key feature of transformer networks is attention [56, 57], more specifically self-attention, which is an operation on a set of elements. Attention allows an element of the set, i.e. a constituent in a jet, to assign weights to other elements. These weights allow the network to determine how much "attention" to assign to different pairs of constituents within each jet during a forward pass. Each constituent in a jet is assigned an attention vector of weights which specifies how much attention to pay to every other constituent in the jet. We use the scaled dot-product multi-headed self-attention of Ref. [58].

To give the reader some intuition for this mechanism we will describe the simpler single-headed attention mechanism, illustrated in Fig. 2. The learnable parameters are contained within just three matrices: the query ($W^Q$), key ($W^K$), and value ($W^V$) matrices. These are learned using standard back-propagation methods in neural network optimization. To explain the mechanism, we start by describing how it is applied it to a single jet constituent described by the embedded phase space coordinate $x_i$ (we slightly abuse our notation such that the index now refers to constituents rather than jets and in the following we will just consider $x_1$ for one of the jets). The query matrix $W^Q$ transforms a single constituent $x_1$ to the corresponding query vector $q_1 = W^Q x_1$. The key matrix $W^K$ then transforms each constituent in the jet to a separate key vector $k_{1,...,C} = W^K x_{1,...,C}$. The query vector $q_1$ is then combined through a scalar product with each key vector and passed to a softmax function, creating the attention weights

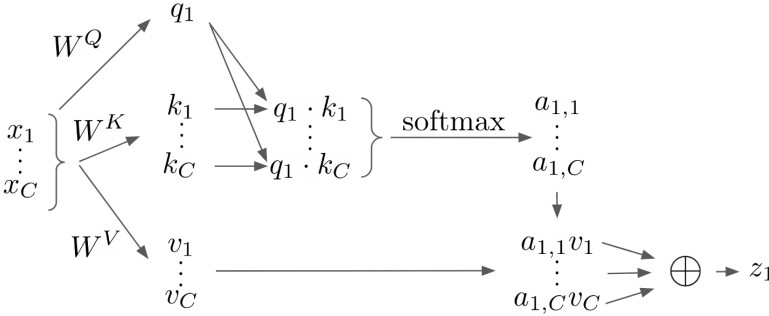

Figure 2: Illustration of scaled dot-product single-headed self-attention. All elements are defined in the text.

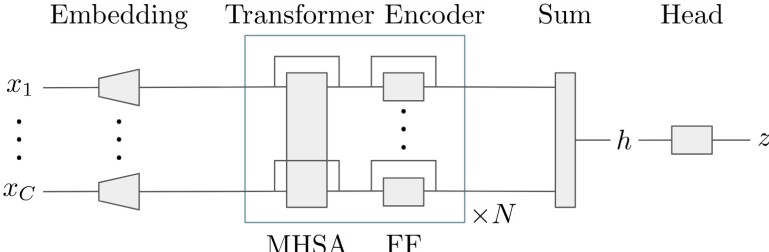

Figure 3: Illustration of the transformer network architecture. MHSA stands for multi-headed self-attention, and FF for a feed-forward block, as defined in the text.

for the first constituent, $a_{1,i} = \mathrm{softmax}((q_1 \cdot k_i)/\sqrt{d})$, where $d$ is the dimension of the query vector. Finally, the value matrix $W^V$ transforms each constituent in the jet to a separate value vector $v_{1,...,C} = W^V x_{1,...,C}$. The output for the first constituent is then just the sum of these vectors weighted by the attention weights, $z_1 = \sum_{i=1,...,C} a_{1,i} v_i$. To summarise, the output for a single constituent is given by

$$z_1 = \sum_i \mathrm{softmax}\left(\frac{q_1 \cdot k_i}{\sqrt{d}}\right) v_i = \sum_i \mathrm{softmax}\left(\frac{(W^Q x_1) \cdot (W^K x_i)}{\sqrt{d}}\right) W^V x_i. \qquad (11)$$

This can be thought of as a projection onto the basis $v_i$, where the coefficients are given by the $\mathrm{softmax}(q_1, k_i)$ between $q_1$ and the $k_i$. The equivalent operation is applied to all $x_j$, leading to a set of outputs $z_j$. Due to the sum over set elements, each output $z_i$ is invariant to the permutation of the other elements of the set, meaning that the entire self-attention operation is permutation equivariant.

A problem with the single-headed self-attention mechanism is that each element of the sequence tends to attend dominantly to itself [58]. This can be solved by extending the network to multiple heads, which we do in the work presented here. This involves several self-attention operations in parallel, each with separate learned weight matrices, then concatenate the outputs before applying a final linear layer. In practice, the full calculation for all constituents, all attention heads, and for an entire batch is carried out in parallel with tensor operations.

**Transformer-encoder network**

In general, transformer networks include a complete encoder-decoder architecture. In our application, we are only interested in deriving a representation $\mathcal{J} \rightarrow \mathcal{R}$, so we use only the encoder part of Ref. [58]. It is a sequence-to-sequence operation, made up of $N$ structurally identical, successive blocks (see Fig. 3).

The starting point is a set of constituents $x_i$, which we embed into a higher-dimensional space by a single learned linear layer without activation. This increases the representational power of the network. A typical dimension of the embedding space is 1000. Working with the embedded jet constituents, each block contains the following operations: multi-headed self-attention (MHSA) is applied to the input constituent, and the result is added to the input, in a residual fashion. This output is normalized using layer normalization [59] and passed through a residual feed-forward (FF) network, which operates on each constituent individually. This transformer-encoder block is repeated $N$ times. Finally, the output is normalized using layer normalization. The encoder outputs are summed over constituents to produce a fixed-size output $h$, which is passed through a final feed-forward head network to give the output $z$. In practice, our supervised linear classifier test will find that $h$ is a better representation than $z$, consistent with typical practice in the self-supervised literature [38]. While the

Table 1: Default setup of the transformer-encoder network and the JetCLR training, unless noted explicitly.

| hyper-parameter | value | hyper-parameter | value |
|---|---|---|---|
| model (embedding) dimension | 1000 | optimizer | Adam ($\beta_1 = 0.9$, $\beta_2 = 0.999$) |
| feed-forward hidden dimension | 1000 | learning rate | $5 \times 10^{-5}$ |
| output dimension | 1000 | batch size | 128 |
| # self-attention heads | 4 | # epochs | 500 |
| # transformer layers ($N$) | 4 | | |
| # layers | 2 | | |
| dropout rate | 0.1 | | |

output of the transformer-encoder is permutation-equivariant, the sum makes the representation $h$ permutation-invariant, similar to the Deep-Sets approach [36,60]. Our network is implemented in PyTorch [61] with the TransformerEncoder module, we also make heavy use of NumPy [62].

**Variable-length inputs**

A general feature of jet constituents is that their number per jet is variable. As in all ML tools for jet analyses, we zero-pad jets with fewer constituents. This makes it easier to convert a batch to a single tensor input for efficient computation and allows us to concatenate the batch elements with equal length. To ensure that this padding does not affect the network output, we implement masking in the transformer. To stop information flow from zero-valued constituents, we require the attention weights corresponding to those constituents to be zero, technically by adding negative infinity to the attention weight before the softmax normalization. In addition, we remove zero constituents from the sum over constituents to ensure that the transformer is completely invariant to zero padding.

This masking ensures that constituents with zero $p_\mathrm{T}$ have no effect on the output, but we can generalize this by defining the masking to be continuous in $p_\mathrm{T}$. Instead of adding negative infinity to some pre-softmax attention weights, we add $\beta \log p_\mathrm{T}$ ($\beta = 0.5$) to all pre-softmax attention weights. In addition, instead of setting some transformer outputs before summation to zero, we multiply all transformer outputs by the input $p_\mathrm{T}$. This IR-safe attention mechanism renders the transformer network IR-safe by construction.

## 4  Pretty good results

After introducing all JetCLR elements, we have to investigate how its various symmetries and augmentations contribute to its performance in top-tagging with a Linear Classifier Test (LCT), and see how our representation compares to alternative approaches. Top-tagging has long been a standard benchmark for testing machine-learning algorithms, building on traditional approaches which for example search for subjets [63] or mass drops [64,65] in the jet substructure. As well as serving as a benchmark, machine-learned top-taggers could play an important role in searches for heavy BSM resonances.

LCTs are widely used in the representation learning literature as a proxy measure for how expressive a representation is. The whole procedure described so far in getting the new representations of the jets using JetCLR is self-supervised, so it does not need to know which jets are QCD and which are top jets. However for the LCT we use the truth labels to train a simple linear classifier to distinguish between QCD and top jets using the representations obtained

Table 2: Left: classification results for JetCLR trained with different symmetries and augmentations and $S/B = 1$. The default setup includes translation and rotation symmetries, combined with soft and collinear augmentations. Right: classification results for the combined (default) symmetries and augmentations, trained with different signal-to-background ratios $S/B$.

| augmentation | $\epsilon_b^{-1}(\epsilon_s=0.5)$ | AUC |
|---|---|---|
| none | 15 | 0.905 |
| translations | 19 | 0.916 |
| rotations | 21 | 0.930 |
| soft+collinear | 89 | 0.970 |
| all combined (default) | 181 | 0.980 |

| $S/B$ | $\epsilon_b^{-1}(\epsilon_s=0.5)$ | AUC |
|---|---|---|
| 1.00 | 181 | 0.980 |
| 0.50 | 160 | 0.979 |
| 0.25 | 150 | 0.978 |
| 0.10 | 161 | 0.978 |
| 0.05 | 146 | 0.978 |
| 0.01 | 158 | 0.978 |

from the JetCLR method. We then do this for other well-known jet representations, and compare their performance to what we get using JetCLR representations.. The idea is that better representations perform better in this test.

Our transformer setup is given in Table 1. The temperature hyper-parameter $\tau$ determines the trade-off between the alignment and uniformity. It can strongly affect the performance of the representations in a LCT. We find that $\tau = 0.1 \ldots 0.2$ works best which, despite the very different applications, is in agreement with the computer vision applications in Refs [38, 66]. We also see that more model dimensions result in better performance, although with the transformer network this performance gain seems to plateau around $1000 \ldots 1500$ dimensions. Earlier tests using a fully-connected network instead of a transformer indicated that this plateau happens at around 200 dimensions. We focus on the 1000-dimensional representations because this will eventually provide a fair comparison to the EFPs at $d \leq 7$, which is also a 1000-dimensional representation of the jets.

Our LCT is a linear neural network with a binary cross-entropy (BCE) loss, optimized using stochastic gradient descent. The network is trained with 50k top and QCD jets each for 5000 epochs with a batch size of 2056. The exact setup along with some alternative LCT setups are discussed in the Appendix, with a focus on their respective strengths and underlying assumptions.

**JetCLR**

We present the results in terms of the Receiver Operating Characteristic (ROC) curve, which is a function of the efficiency ($\epsilon_s$) and the mistag rate ($\epsilon_b$). The efficiency measures the fraction of top jets that pass a cut on the output of the classifier, while the mistag rate measures the number of QCD jets that pass the same cut. The ROC curve we use is simply the function $\epsilon_b^{-1}(\epsilon_s)$ where $0 \leq \epsilon_{s,b} \leq 1$. In an analysis setting one or more working points on this curve would be chosen, and in the tables below we chose the working point corresponding to $\epsilon_s = 0.5$. Note however that in an analysis it would be impossible to know the exact efficiency of the chosen working point. We also quote the Area Under Curve (AUC) to provide an integrated measure of performance, this is defined as $\int_0^1 d\epsilon_b \, \epsilon_s(\epsilon_b)$. This can be interpreted as the probability that a randomly chosen signal jet has a higher output from the classifier than a randomly chosen background event. An uninformative classifier should give an AUC of 0.5 while a perfect classifier should give an AUC of 1.0.

From first principles, it is not clear which symmetries and augmentations work best for learning representations with JetCLR. In the left panel of Table 2 we summarize the results after applying rotational and translational symmetry transformations and soft+collinear aug-

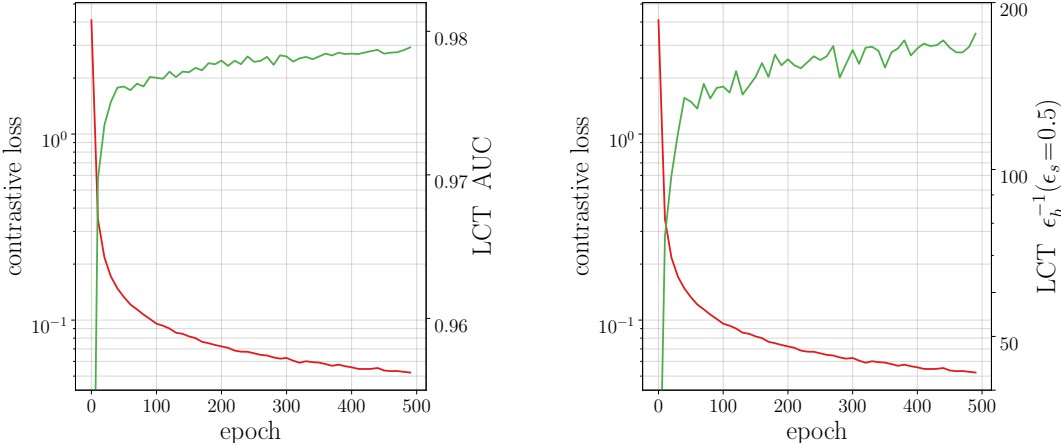

Figure 4: Contrastive loss value and LCT performance on test data as a function of the training epochs. The LCT is performed every 10 epochs.

mentations. To get an idea, we quote the best of a number of runs for each case. Individually, the soft+collinear augmentation works best. Translations and rotations are less powerful individually, but the combination of all three provides by far the best representations. The results for the individual augmentations in Table 2 were obtained using regular masking in the transformer. When combining all symmetries and augmentations the IR-safe masking gives a slight boost, so our default in Table 2 includes IR-safe attention.

While our initial results are based on a dataset containing equal amounts of QCD and top jets, any application to anomaly detection requires our approach to work with much fewer top signal jets. In the right panel of Table 2 we show the performance of our default benchmark for a decreasing fraction of signal events in the training sample. For each signal model with $S/B < 1$ we only train one model, so we expect some noise in the results. The outcome indicates that the JetCLR performance in the LCT is hardly sensitive to the amount of signal jets in the training data, and that JetCLR can encode its fundamental structures based on QCD jets and the symmetries and augmentations alone. Due to the stochastic nature of jet data, no pattern is exclusive to top jets, so the QCD jet sample should indeed contain all relevant information. This result is very promising for future anomaly searches using JetCLR representations.

To test our JetCLR training we analyse the AUC and the mistagging rate of the LCT on the test data as a function of the training epoch. Given the impressive LCT scores we could just assume that optimizing the contrastive loss is a good auxiliary task for constructing good representations for classification. However, for anomaly detection we know that the auxiliary optimization task may appear to be converging to a good representation for classification initially, but then diverge at larger epochs [9, 67]. In Fig. 4 we show that the increasing performance of the JetCLR representations in the LCT is indeed aligned with the optimization of the contrastive loss function.

**Encoded symmetries**

Of the two basic JetCLR tasks, invariance and discriminative power, we first confirm that the network indeed encodes symmetries. To illustrate the encoded rotation symmetry we show how the representation is invariant to actual rotations of jets. We start with a batch of 100 jets, and produce a set of rotated copies for each jet, with rotation angles evenly spaced in $0 \ldots 2\pi$. We then pass each jet and its rotated copy through the network, and calculate their cosine similarity, Eq. (6), with the original jet. In the top panels of Fig. 5 we show the mean and

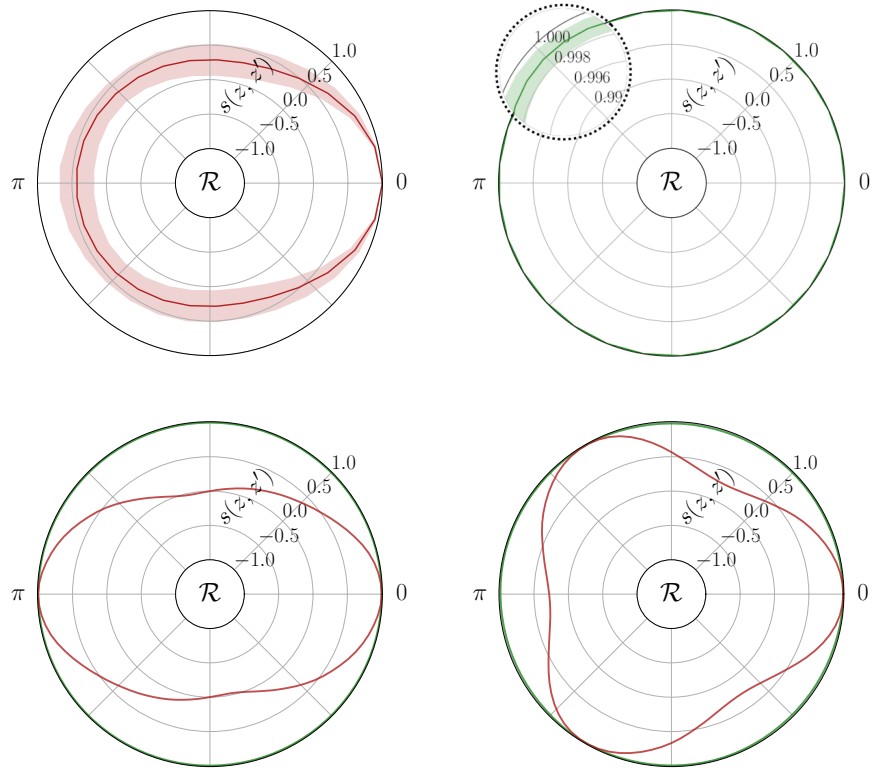

Figure 5: Visualization of the rotational invariance in representation space, keeping in mind that $s(z, z') = 1$ indicates identical representations. Top: JetCLR representation trained without (left) and with (right) rotational transformations. Bottom: JetCLR representation for two-prong (left) and three-prong (right) toy jets, trained without (red) and with (green) rotational transformations.

standard deviation of the cosine similarity as a function of the rotation angle. First, from the scale of the radial axis $s(z, z')$ we see that the representations obtained by training JetCLR with rotations are much more similar to the original jets. Second, in the left panel the similarity varies between 0.5 and 1.0 as a function of the rotation angle, while in the right panel the JetCLR representation is indeed rotationally invariant.

Next, we create toy jets with $p_{\mathrm{T, jet}} = 600\,\mathrm{GeV}$, one with two constituents and one with three equally spaced constituents. The jet momentum is shared equally between the subjets. We then compare how rotationally invariant their JetCLR representations are in the lower panels of Fig. 5. The red lines represent the similarity functions for JetCLR representations of two-prong (left) and three-prong (right) jets, trained without rotational transformations. The maximum values of $s(z, z')$ reflect the degeneracies from the geometric symmetry of the toy jets. The green line represents the similarity function for the JetCLR representations trained with rotational transformations.

**JetCLR performance**

After confirming that the JetCLR indeed encodes symmetries, we turn to the second task, namely discriminative power. To put the results of Table 2 into context, we show ROC curves for JetCLR and various other representations in Fig. 6. For the constituents representation we take the 20 hardest constituents in each jet, flatten their $(p_{\mathrm{T}}, \eta, \phi)$ components into a single vector, and feed them to the linear classifier. For the jet images representation we use the

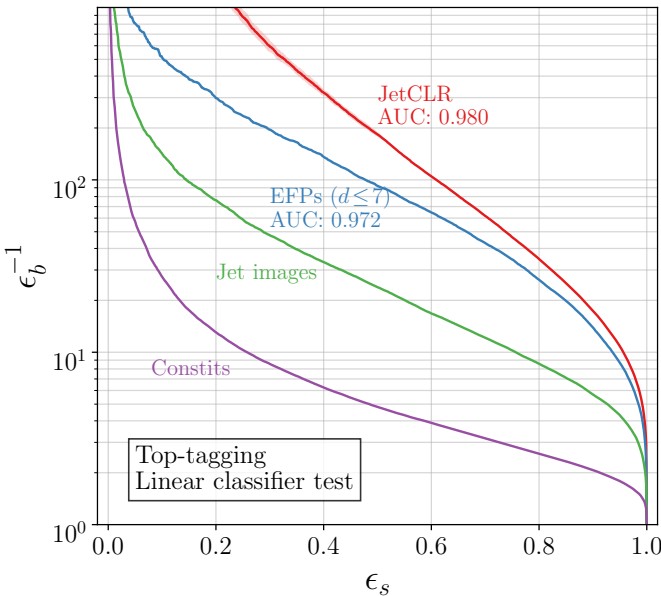

Figure 6: Comparison of JetCLR with other classification metrics.

preprocessing of Refs [6,7], flattening the $40 \times 40$ image to a single 1600-dimensional vector and giving it to the linear classifier. Finally, the EFP representation is invariant to permutations and to rotations by construction, and its IR-safety guarantees independence from soft activity. In many ways, EFPs can be considered a theory-driven counterpart of our JetCLR tool. We use all coefficients up to degree seven, since it has been shown that adding higher powers does not improve the top-tagging [37], and choose $\beta = 0.5$ for the exponent of the $p_T$-weights. The EFPs we used were calculated using the EnergyFlow python package from [37]. For the JetCLR representation we used the default setup, restricting the maximum number of constituents per jet to 50, and masking jets with fewer constituents.

For all representations we train the networks with 100k jets split evenly between top and QCD. For the alternative representations we run eight linear classifiers and use the mean over the mistag rates for the ROC curve. For the JetCLR representation an additional source of uncertainty arises from the training of the transformer-encoder network. We train two linear classifiers on four different representations from four different JetCLR runs and show the mean and standard deviation of the mistag rate vs. the efficiency.

As expected, representations using more knowledge of the physical symmetries perform increasingly well. The top-performing EFP and JetCLR representations use the same latent dimension, and the self-supervised JetCLR method slightly outperforms the systematic EFP representation for a linear network with a binary cross-entropy loss, the LCT with the weakest assumptions about the data. As discussed in the Appendix, the EFP results improve for a linear discriminant analysis, where the underlying assumption on the data is not applicable to JetCLR. The performance of both the EFPs and JetCLR vary depending on the type of linear classifier used. Among the four linear classifiers used the min/max $\epsilon_b^{-1}(\epsilon_s = 0.5)$ performance metric found using EFPs was recorded to be 93/165 while for JetCLR it was found to be 130/181. Of these four linear classifiers the EFPs did outperform JetCLR on one, the linear discriminant analysis. We stress here that we did not do an exhaustive comparison between the different representations. Obviously, any LCT can only serve as a proxy to estimate representation quality, the real test will be performance in an actual analysis.

# 5 Conclusions

We have introduced contrastive learning (CLR) to design observables which respect symmetries and data augmentations while retaining discrimination power within the dataset. We have applied this new method in jet physics, developing the JetCLR tool[1]. Guided by fundamental symmetries and principles of quantum field theory, we introduced a transformer-encoder network to encode rotation, translation, and permutation symmetries, as well as invariance under soft and collinear constituent augmentations.

After visualizing the symmetry-enhanced representation space, we evaluated the network performance using a linear classifier test, a simple supervised classifier trained on the representations to distinguish top jets from QCD jets. Due to the simplicity of the classifier, its performance can be interpreted as a quality measure for the representations. We find that self-supervised JetCLR outperforms simple jet images and is competitive with energy flow polynomials.

Regardless of our specific JetCLR application, our key point is that it is possible to incorporate symmetry principles and physics knowledge in self-supervised ML tools and latent representations. This opens many avenues for future work with JetCLR and contrastive learning in general. Because of the way JetCLR incorporates symmetries from a single augmented data set, it is particularly well suited to enhance and control anomaly searches, one of the great ML opportunities for future LHC runs.

## Acknowledgements

We are, again, grateful to Ullrich Köthe for invaluable discussions. The research of TP is supported by the Deutsche Forschungsgemeinschaft under grant 396021762 – TRR 257 *Particle Physics Phenomenology after the Higgs Discovery*. PS is partly supported by the DFG Research Training Group GK-1940, *Particle Physics Beyond the Standard Model*. BMD is supported by funding from BMBF and the Wissenschaftsministerium Baden-Württemberg through the Excellence Strategy. GK acknowledges the support of the Deutsche Forschungsgemeinschaft under Germany's Excellence Strategy – EXC 2121 *Quantum Universe* – 390833306.

## A Linear classifier tests

Without a direct application to a specific task, comparing data representations is difficult. Downstream tasks can vary from anomaly detection to classification, or regression. One standard method for comparing representations is the linear classifier test (LCT), but even this test can be ambiguous. The idea behind the LCT is that the linearity of the classifier removes much of the expressive power from the classifier, so a linear classifier measures the expressive power of the representation. However, we find that in removing expressive power from the classifier, the results become much more dependent on the inductive biases incurred in the choice of loss function and optimization. We discuss a few different LCTs and explain the assumptions they make about the data they are optimized on. We provide a more complete comparison between the JetCLR and EFP representations using different LCTs in Table 3. All classifiers are trained using 10-fold cross-validation to identify the best-performing hyper-parameters. The reported performance is the average over the 10 folds.

---

[1]The JetCLR code will be maintained at https://github.com/bmdillon/JetCLR.

Table 3: Comparison of JetCLR representations and energy flow polynomials (EFPs) as in Fig. 6, including different linear classifier tests.

| | EFPs ($d \leq 7$) | | JetCLR | |
| --- | --- | --- | --- | --- |
| | $\epsilon_b^{-1}(\epsilon_s = 0.5)$ | AUC | $\epsilon_b^{-1}(\epsilon_s = 0.5)$ | AUC |
| binary cross-entropy (BCE, Fig. 6) | 93 | 0.972 | 181 | 0.980 |
| SVM (hinge loss) | 88 | 0.971 | 130 | 0.977 |
| SVM (squared hinge loss) | 100 | 0.971 | 169 | 0.979 |
| linear discriminant analysis (LDA) | 165 | 0.979 | 133 | 0.977 |

**Binary cross-entropy loss**

A linear classifier trained with binary cross-entropy loss, also known as logistic regression, makes an assumption about how the probability of each of the two classes changes in different parts of the space. If $x$ denotes data and $y \in \{0, 1\}$ the two classes, the assumption is that

$$p(y = 1 \mid x) = \text{sigmoid}(w^T x + c) = \frac{1}{1 + e^{-w^T x - c}}, \tag{12}$$

where $w$ is some vector and $c$ is a scalar bias. We find $w$ and $c$ by minimizing the Kullback-Leibler divergence between this model and the labeled data. In practice, this means minimizing the binary cross-entropy (BCE)

$$\mathcal{L} = \left\langle -\log \text{sigmoid}(y(w^T x + c)) \right\rangle_{x,y} + \lambda \|w\|^2, \tag{13}$$

where $\lambda \geq 0$ is a regularization parameter. Regularization is not strictly necessary but can improve performance. It can be turned off by setting $\lambda = 0$. We select the best regularization parameter $\lambda \in \{10^{-6}, 10^{-4}, 10^{-2}\}$ by 10-fold cross-validation. This optimization problem is convex, so it should always give the same optimal $w$ and $c$ when using a standard algorithm such as gradient descent.

**Support vector machine**

A (linear) support vector machine (SVM) separates two classes with as wide a margin as possible, aiming for robustness. When the two classes are not linearly separable, as in our application, the SVM minimizes how far on the wrong side of the decision boundary misclassified points are, but it does not consider points which are safely on the correct side of the boundary. This is in contrast to logistic regression, which pushes points to the correct side of the decision boundary no matter how far over it they already are. SVMs are expressible as a convex problem and make no assumptions about the distribution of the data. However, they involve tuning a hyper-parameter which determines how strictly misclassifications are enforced. This will lead to different results depending on the choice of hyper-parameter, which must be selected based on performance on some other metric, for instance a classification accuracy. We consider two variants of SVM, starting with the standard hinge loss

$$\mathcal{L} = \left\langle \max(0, 1 - y(w^T x + c)) \right\rangle_{x,y} + \lambda \|w\|^2, \tag{14}$$

where $\lambda > 0$ is the regularization parameter. The variant with the squared hinge loss minimizes

$$\mathcal{L} = \left\langle \max(0, 1 - y(w^T x + c))^2 \right\rangle_{x,y} + \lambda \|w\|^2. \tag{15}$$

The two differ in how strongly they penalize distance from the decision boundary. The squared variant enacts a weaker penalty for points which are only just on the correct side of the boundary, but a stronger penalty for points which are on the incorrect side. We select the best $\lambda \in \{10^{-6}, 10^{-4}, 10^{-2}\}$ by 10-fold cross-validation.

**Linear discriminant analysis**

Linear discriminant analysis (LDA) makes a stronger assumption, namely Gaussian distributed data. The data is modeled as a Gaussian mixture model with two equally likely classes, where the covariance matrix of the two classes is the same. The means $\mu_{0,1}$ and the covariance $\Sigma$ are typically estimated from labeled data. The Bayes-optimal classifier is the log ratio of the two probabilities. Gaussian log probabilities are quadratic in $x$, so if the two covariance terms cancel we obtain a linear equation in $x$. As with the previous classifiers, the classification score can be written as $w^T x + c$, where here $w = (\mu_1 - \mu_0)^T \Sigma^{-1}$ and $c = -w^T(\mu_0 + \mu_1)/2$. No cross-validation is necessary to select hyper-parameters, but the reported results are averages over 10-fold cross-validation. The basic assumption of Gaussian data is not fulfilled by the JetCLR representations, which are optimized for uniformity on a unit hypersphere, so the linear discriminant analysis will not capture the structure of the JetCLR representation.

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
