# Peer review of "Symmetries, Safety, and Self-Supervision"

_SciPost Physics, doi:SciPost Phys. 12, 188 (2022)_

## Round 1 · Referee Report · Anonymous (Referee 1) · 2021-10-28

Strengths

1- novel representation for QCD jets based on recently presented Contrastive Learning

2- technology transfer from self-supervised learning of image representations to HEP

3- potential for future application in data analyses and interpretation

Weaknesses

1- hard to grasp more detailed insights for non expert in ML, similarly for non QCD expert I assume the specifics of jet physics are rather opaque

2- rather cluttered nomenclature in Sec. 3

Report

The paper presents the adaptation and application of the recently presented Contrastive Learning technique to derive suitable representations for QCD jets.
This method has originally been developed for visual representation of actual images. The authors adapt this method to the situation of QCD jets with the particular application example of discriminating plain QCD and top-quark initiated jets in simulated LHC data. However, the scope of the method is wider, including unsupervised anomaly detection in jet data from the LHC experiments.

The authors put particular emphasis on incorporating symmetries relevant for the representation of QCD jets by constructing appropriate augmentations of the jet data used to train the model.

The idea is quite innovative and offers potential for interesting future applications. This certainly qualifies the paper for publication in SciPost. However, prior to publication I would request the authors to address the following list of changes/clarifications.

Requested changes

1- At least a few sentence introducing QCD jets, the central object of the paper, should be added, either in the general intro or in Sec. 2, including what the considered jet constituents represent, i.e. individual particles, detector cells ...

2- Similarly, a brief comment on top-quark vs. light-quark/gluon initiated jets separation seems appropriate.

3- In Sec. 2, when discussing imposed/assumed physical symmetries the made statements should be underpinned by suitable references, in particular about IRC safety.

4- Among the proposed augmentations are translations of the jets in $\eta-\phi$ space, what is the reasoning behind? We know that in particular the $\eta$ distribution of jets is not flat and in fact linked to the jet initiator.

Further, I don't actually understand the relevance of the translations, just before the authors describe that they align each jet with the origin of the $\eta-\phi$ plane, i.e. translating it from its actual position in $\eta-\phi$ anyhow. I am just missing the point?

5- In the intro to Sec. 3 the authors highlight the permutation symmetry of QCD jets, i.e. the invariance under reordering the constituent labels, at least to me the meaning was however not clear in the first go. Maybe a comment would help other readers.

6- The nomenclature used in the paragraph on Attention is quite cluttered and complicated to graps, i.e. when are we considering a single jet, all or just a single jet constituents. This could certainly be improved, e.g. by simply using a second index on the $x_i$. Further, what is $q_i$ standing for beyond $q_1$?

7- On page 9 the authors make a comment about convergence issues in anomaly searches. For me as a non-expert a reference seems to be appropriate.

8- Lastly a more general question, in the way set up the jet representation seems agnostic about the actual production process of jets, e.g. through explicitly imposing rotation symmetry. For certain production modes, however, there might be preferred directions within jets, consider for example jet pull. Would the authors assume that this requires adjustments to the augmentations and correspondingly trainings for specific analyses or to what extend is the proposed setup considered universal?

  • validity: high
  • significance: high
  • originality: high
  • clarity: high
  • formatting: excellent
  • grammar: excellent

Author:  Barry Dillon  on 2022-01-18  [id 2102]

(in reply to Report 1 on 2021-10-28)

We’d like to thank the referee for the constructive and interesting suggestions, I believe we have implemented everything requested by the referee in the resubmitted version.

As for the question in the 8th point in the report, this is very interesting. So far we have only looked in detail at individual jets, so the augmentations are driven towards enforcing symmetries that are completely agnostic to the event-level dynamics. If we were to look at whole events, we would certainly need to reconsider the types of augmentations and symmetries we include. The most obvious of these are the translations in eta. So, the proposed set-up is only universal in the context of studying individual jets. More work needs to be done when applying the technique to multi-jet or whole events.

---

## Round 1 · Referee Report · Anonymous (Referee 3) · 2021-11-16

Report

This is interesting work where the authors aim to exploit intrinsic symmetries of high-dimensional data to discriminate between top quark and QCD jets. They introduce a method called jetCLR (jet observables for Contrastive Learning of Representations) and benchmark it using the linear classifier test. The paper is well written in a clear and didactical style. However, some valid suggestions to improve further the presentation have been made by the other referees. Therefore, I suggest publishing this paper after addressing the points raised in previous referee reports.
  • validity: -
  • significance: -
  • originality: -
  • clarity: -
  • formatting: -
  • grammar: -

Author:  Barry Dillon  on 2022-01-18  [id 2104]

(in reply to Report 3 on 2021-11-16)

Thanks for the comments, we believe we have implemented the suggestions made by the other referees. The details are in the author comments in the resubmission page.

---

## Round 1 · Referee Report · Andy Buckley (Referee 2) · 2021-11-16

Strengths

  1. A useful, and helpfully practical guide to an approach for forcing approximate physical symmetries into neural-net classifiers

  2. Convincing demonstration that this method produces a performant top/QCD jet classifier, comparable to or exceeding the discriminating power of other methods (either fully non-parametric, or using an explicitly physical basis)

Weaknesses

  1. Unclear presentation or assumed knowledge of various ML concepts, particularly performance metrics. Some assumption is reasonable these days, but this paper takes it a little too far, and erects unnecessary barriers to understanding by non-experts. A little text reworking would improve this greatly.

Report

A nice and compact paper, exploring constructively the important business of encoding a priori physical symmetries into neural-network applications in physics. The practical demonstration of methods to force such encoding rather than simply hope for the network to learn known features is as important an output as the JetCLR code itself. I have a few comments primarily on the presention, which is sometimes disjointed or opaque -- particularly for newcomers to the area -- and being more approachable would take little work and be only a good thing.

• Sec 2, p3: Not key to the thrust of the paper, but I assume that (in addition to generation simplicity) the reason to disable MPI as well as ignoring pile-up is that in practice jets would be groomed in some way that effectively removes both contributions? No grooming step is mentioned, but maybe the Delphes particle-flow algorithm is meant to approximate pile-up/UE suppression. Probably the end result is not strongly affected, particularly as only the leading constituents are passed to the network (perhaps better would have been to pre-cluster to a fixed max number of constituents?), but it'd be good to justify the input choices as being reasonably close to the realistic application.

• p3-4: in this presentation it's hard to know what is signposting and what is the actual full presentation of the idea. I think some more advance notice is needed, to paint the picture that what is coming is use of known pairings in the training set, and a matching loss function, to empirically nudge rotational, translational, IRC, and permutation symmetries into the resulting network weights and architecture. It'll then be more obvious what is going on from the start of each subsection, which is currently opaque until a second reading in several cases.

• Sec 3 intro: Maybe note prior art on permutation invariance in the work of Thaler et al on deep sets and energy-flow networks (and EFPs), e.g. https://arxiv.org/abs/1810.05165. This would also clarify what exactly is meant by "permutation symmetry" at the top of p6

• The section on attention is quite opaque to anyone not already familiar with the idea in some depth. The naming of the described operator as "attention" isn't motivated, the distinctions between queries/keys and how their originating W matrices are learned isn't given, and the need for any of this isn't clear other than that the sum over elements makes the network permutation-independent. I feel like this section either needs to be larger and to explicitly motivate this mechanism for learning jet structure beyond being one of several permutation-invariant architectures, or to make it much shorter and simply declare that a self-attention based transformer network provides the desired permutation-independence (intrinsically rather than as a strategy in the training).

• It's nice to have some technical detail on dealing with issues like the variable-length inputs and tweaks to ensure ignoring of the zero-pad elements. Very useful and just the right level, thanks.

• Sec 4: "LCT" is used here for the first time since the single occurence in the introduction: I think it would be better to re-introduce it, and to only define the acronym, at this point where it's used. I'm not entirely unaware of ML terminology, but had to re-search the document to find what was being referred to. Having located it and re-read, I still do not know how the linear classifier cut is related to the trained network -- this basic understanding shouldn't require reading the appendix. The relevance of an LCT, i.e. how the performance is being assessed, should also be made clear here: I don't think enough context was given in the intro, and that was a long time ago, intellectually. The first paragraph of Sec 4 is a very natural place to explain how the testing is to be done. (Minor note, it'd be helpful to give a hint of what the "very different applications" of Refs [37,55] actually are, rather than forcing the reader to jump to the bibliography and cross-check!)

• Tab 2: I know it's standard in ML literature, but AUC (and the epsilons) should be defined, and some motivation given for why it's a good metric (it is not clear to me why the integral performance of the model, rather than the performance of the best working point within it, is the best metric).

• Nice result on the reducing S/B ratio! And also the overall ROC curve look nice -- it would be good, if possible, to comment on why JetCLR appears to be significantly outperforming the EFPs of same latent dimension. Implicit linearity in the discriminator built on EFPs? (I note that little detail is given on the EFP classifier, and it would be good to say more. Also, are the EFPs calculated using the code associated with the original paper?)

Requested changes

  1. Better clarity in the relevance of the detailed description of the self-attention structure: either more or significantly less text, with clear connection to the application.

  2. A more complete description of the role of the LCT in evaluating network performance, without requiring reference to the appendix.

  3. Explicitly define/explain classifier performance metrics and terminology.

  • validity: top
  • significance: high
  • originality: good
  • clarity: good
  • formatting: perfect
  • grammar: excellent

Author:  Barry Dillon  on 2022-01-18  [id 2103]

(in reply to Report 2 by Andy Buckley on 2021-11-16)

We’d like to thank the referee for the constructive and detailed suggestions. We believe we have implemented all of the requested changes/additions in the resubmitted version of the manuscript. Answers to the questions were given in the author comments in the resubmission. If anything is unclear, please let us know.

---

## Round 1 · Referee Report · Anonymous (Referee 4) · 2021-12-5

Report

Very interesting study on the implementation of the already known physical symmetries on the network using contrastive learning. The study has been well documented with exceptions that are already pointed out by other referees. Beyond those comments, I would like to raise two issues.

1) Authors are pointing to a similarity observable to be used in the processing of the network in equation 6. Can they elaborate on this further in terms of, can there be any other approaches such as the usage of different similarity constructions. If so what kind of effect would they create on the outcome of the network. 2) In figure 4 authors are showing the evolution of the test data with respect to each epoch. However, this hasn't been compared to the training data so it's not possible to tell if this model is well trained or not.

  • validity: -
  • significance: -
  • originality: -
  • clarity: -
  • formatting: -
  • grammar: -

Author:  Barry Dillon  on 2022-01-18  [id 2105]

(in reply to Report 4 on 2021-12-05)

We'd like to thank the referee for the comments and suggestions for the manuscript. We have implemented these in the resubmitted draft. The details of the changes are in the author comments in the resubmission page.

---

## Round 2 · Referee Report · Andy Buckley (Referee 2) · 2022-1-19

Strengths

My points from the first reporting round have been implemented satisfactorily, from a quick review of the response and the updated document.

Report

While a good paper, with strong practical advice on this method for encoding physical symmetries via augmentations and permutation invariance, I am not sure its significance meets the criteria given for SciPost Physics: with reference to the criteria, it is not a "groundbreaking discovery" or novel link between research areas, and while an interesting approach with apparent potential, it's a stretch to call this a "breakthrough on a previously-identified and long-standing research stumbling block" or obviously opening "a new pathway"... as perhaps hinted by the (humorous?) conclusion to the abstract. My feeling is that, inasmuch as the distinction matters, according to the strict criteria this would be more appropriate for acceptance without further iteration or review in SciPost Physics Core. From the authors' point of view, I don't think the specific of e-imprint should make any significant practical difference.
  • validity: high
  • significance: high
  • originality: good
  • clarity: high
  • formatting: perfect
  • grammar: excellent

Author:  Barry Dillon  on 2022-01-21  [id 2117]

(in reply to Report 1 by Andy Buckley on 2022-01-19)

While we are grateful to the referee for their comments, we strongly disagree with the referees conclusion that this paper does not meet the standards for SciPost Physics. We address the referees criticisms below:

1 - This paper proposes an entirely new method for training neural networks to have approximate invariance to a pre-defined set of transformations or augmentations of the data. We demonstrated the effectiveness of this method through a comparison with the best and widely used data representations in a linear classifier test. Whether or not this constitutes a "ground-breaking discovery" is of course subjective, but in the application of machine-learning tools to particle phenomenology we certainly feel that it does.

2 - The method builds on cutting edge research on self-supervision from the machine-learning community and is therefore clearly a "novel link between research areas".

3 - Constructing symmetry-invariant representations within deep learning tools has been a "long-standing problem" in particle phenomenology. For example, in the paper we describe the inadequacies of the most widely used representation of jet data in machine-learning applications, the jet images, and how they are preprocessed to achieve rotational and translational invariance. We then explicitly compare our invariant JetCLR representations to the jet images in the linear classifier test.

4 - Given that this is the first paper to propose and demonstrate that self-supervision (in particular constrastive-learning) can be used to enforce approximate invariance to transformations or augmentations of the data in neural networks, this paper clearly opens "a new pathway" for research in machine-learning applications to particle phenomenology problems. While not stated explicitly in the paper, there are many other potential applications of this work in particle phenomenology, and possibly beyond.

5- Our senior author would like to point out that by enforcing the quoted conditions the way the referee is doing it, it is unlikely that ATLAS or CMS would have published any paper since 2012. From the feedback we have received from the theory and experimental community and from our own judgement of conceptual machine learning progress we are convinced that our paper should, without any doubt, be published in SciPost Physics.

---

## Round 2 · Referee Report · Anonymous (Referee 1) · 2022-1-22

Report

I would like to thank the authors for carefully addressing my comments (report 1). I am happy to recommend the publication of the paper in its present form. In contrast to the comment by Andy Buckley, I consider the work quite innovative, in that it transfers and adopts a novel and cutting edge ML method to a challenging and relevant application in HEP, i.e. jet representations. I clearly recommend to publish the article in SciPost Physics!

---

## Round 2 · Author Response

We would like to thank the referees for their comments and suggestions on the manuscript, the resulting paper is now much more accessible and informative.

---

## Round 2 · List of Changes

CHANGES FROM REPORT 1

1 - A sentence has been added in the second paragraph of section 2. A description of what the jet constituents represent has been added to the third paragraph of section 2.

2 - Added a few sentences at the beginning of section 4.

3 - References added on page 5.

4 - While it’s true that the eta distribution of the jets is not flat, the physical properties of the jet, such as mass or subjettiness, are invariant to shifts in eta. Since we look at individual jets alone rather than whole events, we expect the physical properties to only depend on the differences in eta and phi between the constituents within the jet. Much like in jet angularities or the energy flow polynomials.
While applying the augmentations we do centre the jets initially in the eta-phi plane, then perform a rotation around the centre of the jet, and then a random translation in eta-phi. The initial centring is for convenience in doing the rotation, and so that we can can compare the case of using just centred jets vs teaching the network to produce translation invariant representations. A sentence has been added on page 5 to make this clearer.

5 - Two sentences have been added to the beginning of section 3 to clarify this.

6 - The discussion in the “Attention” subsection has been extended and hopefully written in a clearer way. The vector q_i is simply the query matrix multiplied by the vector x_i. This is now clearer from the text.

7 - References added, now on page 10.

8 - This is a very interesting question. So far we have only looked in detail at individual jets, so the augmentations are driven towards enforcing symmetries that are completely agnostic to the event-level dynamics. If we were to look at whole events, we would certainly need to reconsider the types of augmentations and symmetries we include. The most obvious of these are the translations in eta. So, the proposed set-up is only universal in the context of studying individual jets. More work needs to be done when applying the technique to multi-jet or whole events.

CHANGES FROM REPORT 2

1 - Interesting point. The only reason we limit to the 50 hardest constituents is for computational efficiency. We found that once the soft+collinear augmentations and IR-safe masking were added the representations were automatically insensitive to the lower pT constituents. In our application we used the top-tagging dataset from 1902.09914 which is widely used as a benchmark. Here MPI and pile-up are ignored so that the tagging performance and ability to remove MPI/pile-up effects are separated. It would definitely be interesting to use representation learning in the presence of MPI and pile-up, and test ways of removing these effects using self-supervision. A few sentences have been added to page 3 to explain this.

2 - Below eq. 1 we have added a short paragraph to set-up the discussion in this section.

3 - These papers were already cited in the introduction and on page 8, but further citation has now been added to the beginning of section 3. Note that the energy-flow-network citation was accidentally missed in the first arxiv submission, but added for the scipost submission.

4 - The discussion in the “Attention” subsection has been extended and hopefully written in a clearer way. We feel that the attention mechanism deserves some explanation here, so we have attempted to clarify it’s importance in extracting correlations between jet constituents. We have also tried to make the structure of the mechanism clearer in the text, and amended figure 2 slightly such that the indices on the attention vectors are more informative.

5 - Thanks for the comment!

6 - A paragraph and some additional comments have been added to the beginning of section 4. A hint of the “very different applications” is also added.

7 - A paragraph describing the ROC curve is given at the beginning of the JetCLR subsection in section 4.

8 - The EFPs are calculated using the code from the energyflow package presented in the EFP paper, we have now stated this on page 12. It’s already mentioned in the JetCLR performance section that the EFPs perform better using a linear discriminant analysis classifier, with a much more detailed comparison in the appendix. However we have added a few more sentences on this comparison at the end of section 4. We do not comment on the effect that the implicit linearity of the EFPs has on their performance using a linear classifier, only noting that the performance varies a lot depending on the linear classifier used.

The points listed under ‘Requested changes’ are covered in the points above.

CHANGES FROM REPORT 3

None required.

CHANGES FROM REPORT 4

1 - This is an interesting point, however we did not explore alternative distance measures in the representation space. There is a brief discussion on this in the ‘Uniformity vs alignment’ subsection in section 2, and we have added a few sentences here to clarify the role of the distance measure in eq. 6.

2 - In fig. 4 ‘test data’ refers to the data held aside for the linear classifier test, which is used to calculate the performance curves shown in green. This linear classifier test is re-done every 10 epochs, which is the source of the slight variation in the green curve. The red curve however is calculated from the training data used to train the JetCLR transformer, which in turn produces the representations for the test data used in the linear classifier. So the red curve is an indication of how well trained the JetCLR model is, while the green curve is an indication of the quality of the representations from JetCLR at each epoch.

---

## Editorial Decision

published